# Facile Synthesis of Hydrogel-Based Ion-Exchange Resins for Nitrite/Nitrate Removal and Studies of Adsorption Behavior

**DOI:** 10.3390/polym14071442

**Published:** 2022-04-01

**Authors:** Thakshila Nadeeshani Dharmapriya, Hsin-Yin Shih, Po-Jung Huang

**Affiliations:** Institute of Environmental Engineering, National Sun Yat-sen University, Kaohsiung 80432, Taiwan; 93thakshilanadee@gmail.com (T.N.D.); et661155@gmail.com (H.-Y.S.)

**Keywords:** nitrate, nitrite, amine, hydrogel, anion exchange

## Abstract

This research aimed to create facile, reusable, hydrogel-based anion exchange resins that have been modified with two different amines to test their ability to adsorb nitrate and nitrite in water using batch and continuous systems. In the batch experiment, maximum adsorption capacities of nitrate and nitrite onto poly (ethylene glycol) diacrylate methacryloxyethyltrimethyl ammonium chloride (PEGDA-MTAC) and poly (ethylene glycol) diacrylate 2-aminoethyl methacrylate hydrochloride (PEGDA-AMHC) adsorbents can be obtained as 13.51 and 13.16 mg NO_3_^−^-N/g sorbent; and 12.36 and 10.99 mg NO_2_^−^-N/g sorbent respectively through the Langmuir isotherm model. After 15 adsorption/desorption cycles, PEGDA-MTAC and PEGDA-AMHC retained nitrate adsorption efficiencies of 94.71% and 83.02% and nitrite adsorption efficiencies of 97.38% and 81.15% respectively. In a column experiment, modified adsorbents demonstrated adsorption efficiencies greater than 45% after being recycled five times. Proposed hydrogel-based adsorbents can be more effective than several types of carbon-based sorbents for nitrate and nitrite removal in water and have benefits such as reduced waste generation, cost-effectiveness, and a facile synthesis method.

## 1. Introduction

Several nitrogen-containing compounds—including nitrate (NO_3_^−^), nitrite (NO_2_^−^), and ammonium—pollute water resources and can cause severe environmental effects such as eutrophication [1]. Because of its high water solubility, nitrate is potentially the most common groundwater contaminant in the world, posing a serious threat to drinking water supplies [2,3]. The main sources of nitrate and nitrite exposure to the general population are through ingestion of food and drinking water; approximately 5% to 8% of ingested NO_3_^−^ is reduced to NO_2_^−^ by bacteria in the mouth [4]. Increased nitrate and nitrite concentrations in drinking water have detrimental health effects in humans. A common example is the induction of ‘blue-baby syndrome’ (methemoglobinemia), particularly in infants, which occurs when NO_3_^−^ in drinking water is converted into NO_2_^−^ by bacteria and formed NO_2_^−^ that oxidizes the ferrous ion in hemoglobin to ferric ions and then methemoglobin (MetHb) is formed. MetHb cannot carry oxygen and the condition of methemoglobinemia is characterized by cyanosis, stupor, and cerebral anoxia [4,5,6,7,8]. Other health impacts of nitrate and nitrite are gastrointestinal tract tumors, urinary tract and brain tumors, and non-Hodgkin’s lymphoma (NHL) [4]. The main point and non-point sources of nitrate and nitrite contamination in groundwater are the intensive use of chemical fertilizers and manure in agriculture, urban runoff, unsafe disposal of untreated sanitary and industrial wastes, leakage from septic systems, animal manure, landfill leachate, and inadequate wastewater treatment and collection systems [2,4,9,10,11]. As excess NO_3_^−^ and NO_2_^−^ concentrations in drinking water can cause health problems, various environmental regulatory agencies have announced permissible maximum contaminant levels (MCLs) of nitrate and nitrite in drinking water of 10 mg/L (NO_3_^−^) and 1 mg/L (NO_2_^−^) respectively as determined by the United States Environmental Protection Agency (U.S. EPA) [12,13].

Various physical, chemical, and biological methods have been applied for removing nitrate and nitrite in water. Conventional techniques such as adsorption [14,15,16,17,18,19], reverse osmosis [20], ion exchange [21,22,23], zero-valent iron (Fe^0^) [24,25,26,27,28], zero-valent magnesium (Mg^0^) [29], electro dialysis [30], catalytic denitrification [31], and biological denitrification [32] are the most commonly used treatment methods to remove/reduce NO_3_^−^ in recent decades [1,2]. However, current nitrate and nitrite removal technologies have their own strengths and limitations, and have been found to be expensive and less effective, in addition to generating additional by products [2]. Among them, adsorption is regarded as one of the most effective methods for removing nitrate and nitrite due to its convenience in synthesis of adsorbent materials with high efficiency and selectivity. In addition, in the economic and environmental aspects, they are inexpensive and eco-friendly [2,33,34]. Variety of adsorbents—including carbon-based sorbents [35], natural sorbents [36], agricultural waste as sorbents [19,37,38], industrial waste [39], and miscellaneous sorbents [40,41,42] as sorbents—have been developed and tested to adsorb nitrate and nitrite. Adsorbates are physically or chemically attached to the adsorbent’s surface and then adsorbed primarily through ion exchange, coordination interaction, electrostatic interaction, physical adsorption, or chemical interaction [34,43].

Because of its simplicity, selectivity, effectiveness, recovery, and low cost, the ion exchange process appears to be the best choice for small water suppliers contaminated by nitrate [22,44]. Several nitrate selective resins have recently been developed. Nitrate selective resin has shown a decreasing affinity for the following ions: NO_3_^−^ > PO_3_^2−^ > NO_2_^−^ > Cl^−^ > HCO_3_^−^ > OH^−^ > [22,45,46]. The ion exchange process involves passing nitrate/nitrite-loaded water through a resin bed containing strong base anion exchange resins, where nitrate/nitrite ions are exchanged for chloride until the exchange capacity of the resin is exhausted. A concentrated solution of sodium chloride is used to regenerate the exhausted resin [22].

Hydrogels are hydrophilic polymers that are swollen by water and insoluble owing to physical or chemical cross-links. These are widely used as biomaterials for complex device fabrication, cell culture for tissue regeneration, and targeted drug release in biomedical applications. Polyethylene glycol (PEG) is a non-toxic, non-immunogenic, biocompatible, hydrophilic hydrogel, and PEG derivatives—such as polyethylene glycol diacrylate—are most commonly functionalized with vinyl groups at the chain ends (PEGDA). PEGDA could photo-cross-link to form hydrogel adsorbents with biodegradable and high swelling characteristics that could be applied in recovery of heavy metals in waste waters and adsorption of dyes [47,48,49,50].

This work was aimed at the synthesis of facile, reusable, hydrogel-based anion exchange resin modified with two different amines to explore its ability to remove nitrate and nitrite in water using both batch and continuous systems. PEGDA was the hydrogel that acts as the main body while the methacryloxyethyltrimethyl ammonium chloride (MTAC) and 2-aminoethyl methacrylate hydrochloride (AMHC) were the modifiers used in the synthesis of the anion exchange resins. MTAC was chosen because several nitrate and nitrite specific strong base anion resins were recently functionalized by reaction with trimethylamine to create quaternary ammonium exchange sites with high affinity for nitrate and nitrite, whereas AMHC was chosen to compare the nitrate and nitrite removal efficiencies of resins modified with primary and quaternary amines [22,51]. The characteristics of the hydrogel adsorbents and the adsorbate were explored through the adsorption kinetic model and the isothermal adsorption model, and the effect of different environmental parameters on the adsorption efficiency was investigated through the thermodynamic parameters. In addition to adsorption benefits, desorption efficiency was also one of the main purposes of this study.

## 2. Materials and Methods

### 2.1. Materials

Poly (ethylene glycol) diacrylate (PEGDA) (Mw: 700) and 1-Vinyl-2-pyrrolidone were purchased from Aldrich. 2-aminoethyl methacrylate hydrochloride (AMHC) and 2,2-dimethoxy-2-phenylacetophenone were purchased from Acros. Methacryloxyethyltrimethyl ammonium chloride (MTAC) was purchased from Alfa Aesar. Analytical grade KNO_3_ and NaNO_2_ were used as the source of nitrate and nitrite anions respectively. NaOH, HCl, and KCl were also of analytical grade. A stock solution of nitrate and nitrite from KNO_3_ and NaNO_2_ was prepared by dissolving an appropriate amount of matter in deionized water.

### 2.2. Preparation of PEGDA/MTAC and PEGDA/AMHC Composites

PEGDA was added to MTAC (very short form = M) and AMHC (very short form = N) solution, wherein PEGDA/MTAC and PEGDA/AMHC mole ratios were 1:0.5 (M/N 0.5), 1:1.0 (M/N 1.0), 1:1.5 (M/N 1.5), and 1:2.0 (M/N 2.0), dissolved in 1 mL of distilled water. Then 30% or 3% weight per volume of photoinitiator (2,2,-Dimethoxy-2-phenylacetophenone was dissolved in 1-vinyl-2-pyrrolidone) was added to the solution and mixed by a vortex stirrer. The obtained mixture was injected into 2 × 2 × 2 mm molds, and then exposed to UV light (λ ≈ 365 nm) from a 100 W mercury lamp for 3 min or 10 min at room temperature, wherein radical polymerization of PEGDA took place, obtaining light-yellow poly(PEGDA)-based hydrogels. The product was washed 3 times with distilled water to remove the residual chemicals, dried at 60 °C overnight, and then used in the following characteristics and adsorption tests. The preparation process of PEGDA-MTAC/AMHC is shown in Figure 1.

### 2.3. Characteristics Analysis

The physiochemical properties of the PEGDA/MTAC and PEGDA/AMHC resins were determined by a zeta potential analyzer (Zeta Plus, Brookhaven Instruments Corporation, 750 Blue Point Road, Holtsvile, New York, USA) and FT-IR analyzer. The samples were prepared in 30 mL of 0.001 M KCl solution containing 0.015 g of ground PEGDA/MTAC and PEGDA/AMHC resins. Each sample powder (0.015 g) was dispersed in 30 mL of 0.001 M KCl solution by ultrasonication for 30 min. The zeta potential of prepared samples was determined in triplicate by testing the suspension with the initial pH value. To investigate the isoelectric point, the pH of the suspension was adjusted to range of 2 to 11 by adding 0.25 M HCl or NaOH solution. At each pH value, three zeta potential readings were taken, and the average zeta potential at that pH value was plotted against the pH value. The IEP was determined by taking the pH value at which the zeta potential was zero [52]. The functional groups presenting in PEGDA/MTAC and PEGDA/AMHC resins were determined by the FTIR spectroscopy (Nicolet iS50, Thermo Fisher Scientific, 5225 Verona Road, Madison WI 53711-4495, USA). The spectrum was scanned from 4000 to 400 cm^−1^, the resolution was 4 cm^−1^, and the number of scans was 32 times. 

### 2.4. Batch Experiments

The remaining concentration in the supernatant solution after the adsorption process was analyzed spectrophotometrically (nitrate was analyzed by ultraviolet spectrophotometric screening method in which the absorbance was measured at the wavelength of 220 nm and 275 nm. Nitrite was analyzed by the colorimetric method in which the absorbance was measured at the wavelength of 543 nm). For the best adsorption processes; the effect of process parameters—such as equilibration time, pH (adjusted to a set value using HCl and NaOH solutions), MTAC or AMHC concentrations, adsorbate concentrations, and temperature—were determined by changing one parameter and keeping the other parameters constant. The adsorption capacity was calculated using the following Equation (1)
(1)$$q_{e}=\frac{\left(C_{0}-C_{e}\right)V}{W}$$
where q_e_ is the adsorption capacity (mg/g), C_0_ is the initial concentration of anion (mg/L), C_e_ is the concentration of anion in solution at equilibrium time (mg/L), V is the solution volume (L), and W is the adsorbent weight (g).

#### 2.4.1. Batch Experiment I

Batch adsorption studies were carried out in a bottle containing 10 mL of nitrate and nitrite solutions (50 mg N/L) and 0.04 g of adsorbent at different pH. Then the reaction mixture was stirred in a horizontal shaker (110 rpm) at room temperature for a specific time followed by filtration. Regeneration of hydrogels after the adsorption studies of both nitrate and nitrite were carried out using 26.45 wt %, 0.8 wt %, and 0.2 wt % of NaCl prior to next use. The efficiency of the regenerated samples was checked up to the 15th cycle.

#### 2.4.2. Batch Experiment II

Batch adsorption studies were carried out in a beaker containing 100 mL of nitrate and nitrite solutions (50 mg N/L) and 0.4 g of adsorbent at normal pH. Then the reaction mixture was stirred in a magnetic stirrer (360 rpm) at different temperatures (293 K, 303 K, 313 K, 323 K, and 333 K) for a specific time followed by filtration. Following adsorption of both nitrate and nitrite, regeneration of hydrogels was carried out using 26.45 wt %, 0.8 wt %, and 0.2 wt % NaCl at various contact times. The released concentrations of the nitrate and nitrite were measured using an ion chromatograph (Eco IC, Metrohm., Switzerland).

### 2.5. Column Experiments

Column studies (down-flow mode) were carried out in a glass column with an internal diameter of 1.5 cm and 20 cm in length at room temperature. The flow rate was constant with 3 mL/min and the concentration of nitrate and nitrite solutions was 50 mg N/L. The packed-out M 1.5 in the column was 3.0 g. The effluent solutions were collected over a period of 18 h to determine the residue concentrations in the effluent solutions. The column flow was continued until the effluent concentration (C_t_) approached the influent concentration (C_0_), C_t_/C_0_ = 0.95. The regeneration of saturated M 1.5 was carried out using 500 mL of 0.8 wt % NaCl solution (co-flow mode) and the flow rate was constant with 10 mL/min after being washed with distilled water. The efficiency of the regenerated samples was checked up to the 5th cycle. The value of the total mass of anion adsorbed on to the M 1.5 can be calculated using Equation (2). Equilibrium anion uptake or maximum capacity of the column $q_{e\left(\exp\right)}$ is calculated using Equation (3). Influent total anion mass of the column is calculated from the Equation (4) and the removal percentage of anion is obtained from Equation (5).
(2)$$q_{0.95}=\frac{Q}{1000}\int_{0}^{t_{0.95}}(C_{0}-C_{t})\;\mathrm{dt}$$
(3)$$q_{e\left(\exp\right)}=\frac{q_{0.95}}{X}$$
(4)$$W_{\mathrm{total}}=\frac{C_{0}t_{0.95}}{1000}$$
(5)$$\mathrm{Removal}\left(\%\right)=\frac{q_{0.95}}{W_{\mathrm{total}}}100\%$$
where q_0.95_ is C_t_/C_0_ = 0.95 total adsorption mass (mg), C_0_ is the initial concentration of anion (mg/L), C_t_ is the concentration of anion in solution at a set time (mg/L), Q is the flow rate (mL/min), t_0.95_ is C_t_/C_0_ = 0.95 required time (min), X is the adsorbent weight (g), q_e(exp)_ is the experiment total adsorption capacity (mg/g), and W_total_ is the influent total mass (mg).

## 3. Results and Discussions

### 3.1. Characteristics Analysis

#### 3.1.1. Zeta Potential Analysis

The zeta potential of PEGDA-MTAC and PEGDA-AMHC as a function of pH were shown in Figure 2. When the initial pH of the solution increases from 2.0 to 12.0 the zeta potential of PEGDA-MTAC and PEGDA-AMHC were from +6.91 to −13.12 mV and from +3.86 to −16.57 mV respectively. The isoelectric point (IEP) for PEGDA-MTAC was 10.26; however, the IEP for PEGDA-AMHC was 10.22. These results emphasized that large amounts of negative potential occurred at the adsorbent when the initial pH was higher than 10 [52].

#### 3.1.2. FT-IR Spectra Analysis

Figure 3 shows the FT-IR spectra of raw PEGDA (black line), PEGDA-MTAC^a^/AMHC^b^ (red line), and after the nitrate (blue line) and nitrite (green line) adsorbed onto PEGDA-MTAC/AMHC. FTIR analyses and peak assignments of both unloaded and loaded hydrogels are represented in Table 1. The adsorption band at 3432 cm^−1^ was due to the O–H stretching of internal water in the hydrogel [33]. The adsorption band at 2873 cm^−1^ was ascribed to the stretching vibrations of C-H in methylene and methyl groups of PEGDA and PEGDA-MTAC/AMHC. The sharp peaks at 1732 cm^−1^ and 1683 cm^−1^ were attributed to C=O stretching of ester groups in PEGDA and PEGDA-MTAC/AMHC respectively [53]. The broad peak at 1107 cm^−1^ was due to the C–O stretching vibration of ester groups of in PEGDA and PEGDA-MTAC/AMHC [53,54]. The modified adsorbent PEGDA-MTAC/AMHC showed the corresponding peaks of pure PEGDA and MTAC/AMHC. The peaks at 1466 cm^−1^ and 1266 cm^−1^ were due to the C–H bending and C–N stretching respectively in PEGDA-MTAC/AMHC respectively. The peak at 1387 cm^−1^ was attributed to the N–O stretching of the nitrate adsorbed PEGDA-MTAC/AMHC [55,56]. According to the results of the above analysis, the characteristic signals of PEGDA, MTAC, and AMHC can be found on the FT-IR spectrum of PEGDA-MTAC and PEGDA-AMHC modified hydrogels, confirming that the hydrogels were successfully modified and synthesized. Additionally, it confirmed that nitrate and nitrite have been adsorbed onto the PEGDA-MTAC/AMHC.

#### 3.1.3. Effect of Modifier Dose

A chelating resin that can be used as an adsorbent is made up of two parts: the functional group (modifier) that forms the chelate interaction with ions and the polymeric matrix that serves as the support [33]. The effect of modifier dose on the adsorption capacity of adsorbent was studied using a different modifier dose at a fixed initial nitrate/nitrite concentration, pH, temperature, and adsorbent dose. Figure 4 depicts the effect of the modifier dose on the adsorption of nitrate and nitrite at a fixed adsorbent dose of 4 g/L. According to the results obtained from Figure 4, both PEGDA-MTAC and PEGDA-AMHC illustrated similar trends for adsorption of nitrate and nitrite. The adsorption capacity was increased significantly with the elevating modifier dose which was obvious due to the exchange of nitrate or nitrite with existing Cl^−^ on the adsorbent surface being increased with the modifier dose [1,22]. However, when the mole ratios of PEGDA-MTAC were 1.5 and 2.0, the nitrate adsorption capacities were similar. This can be attributed to the limitation of the modified hydrogel dose per unit of volume.

#### 3.1.4. Effect of pH

One of the most important parameters that determines the efficiency of an ion exchange process is the pH of the solution. It also influences the affinity of the contaminant ions for the binding sites of the resin [52]. The effects of pH on the adsorption of nitrate and nitrite were determined at different pH levels on both PEGDA-MTAC (M 1.5) and PEGDA-AMHC (N 1.5) and are presented in Figure 5. The results are shown in Figure 5a had indicated an increasing trend in the nitrate adsorption capacity when the pH of the medium increased from 2 to 10. However, the nitrate adsorption capacity was drastically decreased when the pH of the medium was higher than 11. Figure 5b presents the effect of different pHs on the adsorption capacity of M 1.5 and N 1.5 for nitrite. The results had emphasized a relatively constant trend in the nitrite adsorption capacity from pH 5 to 10. The adsorption capacity was significantly decreased with the further increase in pH from 11 to 12. This is because when the negatively charged surface of the adsorbents is at a pH above the isoelectric point, the negatively charged nitrate and nitrite ions are not attracted to the surface of the PEGDA-MTAC/AMHC resins. Because of the increased negatively charged surface sites, the repulsive electrostatic interaction between the surface of adsorbent and nitrate or nitrite ions would be enhanced at alkaline conditions. On the contrary, the rates of nitrate and nitrite ion sorption are enhanced when the pH values are lower than the isoelectric point of the surface of absorbents [52,57,58]. Furthermore, the presence of excess OH^−^ ions at higher pH values would compete with nitrate or nitrite for adsorption sites on the adsorbent, resulting in a decrease in nitrate/nitrite adsorption capacity [46]. Thus, the adsorption mechanism is not only an electrostatic attraction but also an ion exchange.

#### 3.1.5. Effect of Temperature

The effect of adsorption equilibrium time for adsorption of nitrate and nitrite on to PEGDA-MTAC/AMHC at different temperatures (293 K, 303 K, 313 K, 323 K, and 333 K) are illustrated in Figure 6 and Figure 7. The times for adsorption equilibrium at low (293–313 K) and high (323 K, 333 K) temperatures were observed as 45 min and 30 min respectively. The reason behind this scenario was the increment of reaction rate with the temperature. This can be attributed to the high temperature enhancing the average kinetic energy and effective collision frequency of the molecules.

#### 3.1.6. Adsorption Kinetics and Thermodynamics

To investigate the adsorption mechanism and to realize the potential rate-controlling steps of nitrate and nitrite adsorption onto the PEGDA-MTAC/AMHC, three kinetics models including pseudo-first-order, pseudo-second-order, and intraparticle diffusion kinetics model were employed. Pseudo-first-order model is used for the sorption of solid/liquid systems and it can be expressed as Equation (6),
(6)$$\begin{array}{c} \log\left(q_{e}-q_{t}\right)=\log q_{e}-\frac{k_{1}}{2.303}t \end{array}$$
where q_e_ (mg/g) is the equilibrium adsorption capacity of nitrate or nitrite ion and qt (mg/g) is the amount of adsorbed nitrate or nitrite ion at adsorption time t (min), k_1_ (min^−1^) is the rate constant of pseudo-first-order adsorption. The straight lines in the linear plots of log (q_e_ − q_t_) against t indicate the applicability of the pseudo-first-order model. The slope of the straight-line plot of log (q_e_ − q_t_) against t gives the values of the pseudo-first-order rate constant (k_1_) and r^2^ present in Table 2 and Table 3 [1,59].

The pseudo-second-order model was developed by Ho and McKay [60] firstly, assuming that the chemical sorption is the rate-limiting step. Its integration form can be expressed as Equation (7),
(7)$$\frac{t}{q_{t}}=\frac{1}{k_{2}q_{e}{}^{2}}+\frac{1}{q_{e}}t$$
where q_e_ and q_t_ are the adsorption capacity at equilibrium and at time t (mg/g) respectively, k_2_ is the rate constant of pseudo-second-order (g/mg·min). The value of r^2^, q_e_ (1/slope), k_2_q_e_^2^ (1/intercept), and k_2_ (slope^2^/intercept) of the pseudo-second-order equation can be found out by plotting t/q_t_ against t which are shown in Table 2 and Table 3 [1,33,61].

In the liquid–solid system, the diffusion rate of nitrate or nitrite was determined using the intraparticle diffusion model which was proposed by Weber and Morris [62] and McKay and Poots [63]. The linearized equation can be expressed as Equation (8),
(8)$$q_{t}=k_{p}t^{0.5}+C$$
where k_p_ is the intraparticle diffusion coefficient (mg/g min^0.5^), and C is the intercept which is constant for the experiment [64]. The values of the intraparticle diffusion coefficient and r^2^ can be found by plotting q_t_ against t^0.5^ and are shown in Table 1 and Table 2.

The values of q_e_ were decreased with the increasing temperature of the medium, indicating that the adsorption rates of nitrite and nitrate were inversely proportionate with the temperature. According to the fitted data, the values of pseudo-second-order rate constant k_2_ were raised parallel to temperature. When the values of correlation coefficients of the pseudo-first-order, pseudo-second-order, and intraparticle diffusion models are compared, the r^2^ values for the pseudo-second-order model are greater than the r^2^ values for the pseudo-first-order and intraparticle diffusion models, as shown in Table 2 and Table 3. These results indicate that the kinetic modeling of nitrate and nitrite adsorption onto adsorbent in this work showed better agreement with the pseudo-second-order model with correlation coefficients r^2^ ranging between 0.98 and 0.99 than the other two models. Hence, the kinetics of nitrate and nitrite adsorption onto the PEGDA-MTAC/AMHC were attributed to chemical sorption [61].

The activation energies for nitrate and nitrite adsorption onto the PEGDA-MTAC/AMHC were examined by applying the Arrhenius equation on the pseudo-second-order rate constant. The Arrhenius equation is indicated as
(9)$$k_{2}={\mathrm{Ae}}^{-\frac{E_{a}}{\mathrm{RT}}}$$

The linear form of the Arrhenius equation is
(10)$${\mathrm{lnk}}_{2}=\mathrm{lnA}-\frac{E_{a}}{\mathrm{RT}}$$
where A is the pre-factor in the Arrhenius equation (mg/g min), where k_2_ (mg/g min) is the rate constant of the pseudo-second-order adsorption kinetic at temperature T (K), R is the universal gas constant (8.314 J/mol K), T is the absolute temperature (K), E_a_ (kJ mol^−1^) is the activation energy of the adsorption reaction. The magnitude of the activation energy can provide information about the type of sorption. The E_a_ and A were calculated using the slope and intercept respectively of the plot of ln k_2_ values versus 1/T using Equation (10) [65,66].

Figure 6 and Figure 7 reveal the information related to the fluctuation of nitrate and nitrite adsorption rates with the temperature. The calculated values of activation energies and the frequency factors for nitrate and nitrite adsorption were listed in Table 4. In general, a low E_a_ value (<42 kJ/mol) denotes a diffusion-controlled process, where a high E_a_ value (>42 kJ/mol) denotes a chemically controlled process [66]. The values of E_a_ were laid between 25.13 and 29.81 kJ/mol for the reaction of nitrate and nitrite adsorption onto adsorbents used in this study, indicating that the rate-determining step would be a diffusion-controlled process.

Thermodynamic parameters associated with adsorption were calculated as follows: standard free energy change (ΔG^0^), standard enthalpy change (ΔH^0^), and standard entropy change (ΔS^0^). Equation (11) gives the free energy of the adsorption process when the adsorption equilibrium coefficient K_0_ is taken into account [1].
(11)$$\Delta G^{0}=-{\mathrm{RTlnK}}_{0}$$

The values of enthalpy (ΔH^0^) and entropy (ΔS^0^) were respectively determined from the slope and intercept of the linear van’t Hoff plot (Equation (12)).
(12)$${\mathrm{lnK}}_{O}=-\frac{1}{T}\frac{\Delta H^{0}}{R}+\frac{\Delta S^{0}}{R}$$
where ΔG^0^ is the standard free energy of adsorption (kJ/mol), R is the universal gas constant (8.314 J/mol K), T is the absolute temperature (K), K_0_ is the adsorption distribution coefficient (L/mol), ΔH^0^ is the standard enthalpy change (kJ/mol), and ΔS^0^ is the standard entropy change (kJ/mol K).

The calculated values are illustrated in Table 5 and Table 6. The positive values of ΔH^0^ indicate that the adsorption process of nitrate and nitrite is an endothermic reaction. The negative values of ΔS^0^ express that the randomness was decreased at the solid and solution interfaces during nitrate and nitrite adsorption. Furthermore, it was indicated that the reaction was a non-spontaneous reaction. The positive values of ΔG^0^ express the non-spontaneous reaction for nitrate and nitrite adsorption onto M 1.5 and N 1.5 [1,65,66]. In addition, the ΔG^0^ was increased with the increasing temperature leading the reaction trend to non-spontaneous reaction. These results confirm that the adsorption of nitrate and nitrite onto M 1.5 and N 1.5 in high temperatures is difficult.

#### 3.1.7. Adsorption Isotherms

At a certain temperature, the relationship between the adsorption capacity of the adsorbents and equilibrium concentrations of the adsorbents in the solution were shown in Figure 7. The most commonly used Langmuir model was applied in this study to interpret the adsorption experimental data, and the data were shown in Table 5 and Table 6. Equation (13) expresses the linearized form of the Langmuir isotherm model as
(13)$$\frac{C_{e}}{q_{e}}=\frac{1}{q_{m}K_{L}}+\frac{C_{e}}{q_{m}}$$
where C_e_ is the equilibrium nitrate/nitrite concentration in the solution (mg/L), and q_e_ is the equilibrium nitrate/nitrite concentration on the adsorbent (mg/g). q_m_ is the monolayer capacity of the adsorbent (mg/g) and K_L_ is the Langmuir adsorption constant (L/mg) while q_m_ and K_L_ were calculated from the slope and intercept of C_e_/q_e_ against C_e_ plot [65].

According to the Langmuir model, monolayer adsorption occurs on a homogeneous adsorbent surface, and the adsorption energy of all active sites on the adsorbent is always similar [33]. Table 5 and Table 6 depicted the fitting experimental data of nitrate and nitrite adsorption on to the PEGDA-MTAC/AMHC for the Langmuir isotherm. The q_m_ values of M 1.5 and N 1.5 for adsorption of nitrate at 293 K were 13.66 mg/g and 11.25 mg/g respectively. As well as q_m_ values of M 1.5 and N 1.5 for adsorption of nitrite at 293 K were 13.30 mg/g and 13.12 mg/g respectively. The correlation coefficients values of both M1.5 and N 1.5 exhibit almost perfect agreement between the experimental data and the Langmuir model. This observation suggests that the nitrate and nitrite adsorption by these absorbents was monolayer-type and consistent with the general observation that the adsorption occurs in an aqueous medium usually forms a layer on the adsorbent surface. In addition, it was observed that the adsorption capacities of M 1.5 and N 1.5 for nitrate and nitrite were decreased with the increasing temperature. This might be attributed to the increasing temperature leading the reaction trend to become a non-spontaneous reaction. It can also be observed that the adsorption capacity of nitrate is higher than that of nitrite for both adsorbents. This is due to the fact that nitrate has a greater affinity for surface functional groups than nitrite. In 2014, Sowmya et al. developed a novel chitosan–melamine–glutaraldehyde resin which was quaternized with glycidyl glycidyl trimethyl ammonium chloride for the removal of nitrate and phosphate anions. Investigations revealed that the nitrate and phosphate adsorption capacities of the resin were 97.5 and 112.5 mg/g respectively. Two different adsorption capacities were observed due to the different affinities of two anions towards the surface functional groups of the resin [67].

A number of adsorbents have been used to eliminate nitrate from water and a comparative account as per the literature is shown in Table 7. Nitrate adsorption capacities of activated carbon adsorbents—such as ZnCl_2_ treated coconut granular activated carbon, modified lignite granular activated carbon, untreated coconut granular activated carbon, wheat straw charcoal, and cross-linked and quaternized Chinese reed—were lower than the PEGDA-MTAC/AMHC. Carbon-based materials are known having tremendous adsorption capacity due to their high specific surface area. Activated carbon (AC) is the mostly used for adsorption purpose. Even though activated carbon exhibited high surface area, surface hydrophilicity would limit the adsorption behavior. Surface modification of AC is required to improve the sorption capacity. However, adsorbents such as polyvinyl alcohol/chitosan and calcined (Mg–Al) hydrotalcite exhibited higher adsorption capacities than this study. HDTMA modified QLD-bentonite showed almost similar sorption as PEGDA-MTAC/AMHC.

### 3.2. Reuse of the Adsorbent

If an adsorbent cannot be effectively desorbed or regenerated, its application value is alleviated, and it can influence secondary environmental pollution. Hence, the desorption and regeneration studies of M 1.5 and N 1.5 were carried out and the results are shown in Figure 8 and Figure 9. Figure 8 represents the time course for desorption of nitrate and nitrite from spent M 1.5 and N 1.5 using different concentrations (0.2 wt %, 0.8 wt %, and 26.45 wt %) of NaCl solution as the desorbing eluent at room temperature. Both composites achieved the desorption equilibrium within 15 min for desorption of nitrate and nitrite. The effect of desorption for nitrate followed the sequence as 26.45 wt % > 0.80 wt % > 0.20 wt % of NaCl concentrations while the effect of desorption for nitrate followed the sequence as 0.80 wt % > 0.20 wt % > 26.45 wt % of NaCl concentrations. According to the experimental results, when the concentration of NaCl was increased, the desorption effect of nitrate also increased and it is true for the desorption effect for nitrite; however, the desorption was not effective in 26.45 wt % of NaCl. The reason for this might be the affinity of nitrite for composites is weaker than nitrate hence too many Cl^−^ ions were not required for the desorption process. Then the adsorption/desorption processes were repeated with the regenerated adsorbent to determine the adsorption efficiencies of nitrate and nitrite in each repeated use as shown in Figure 9. Adsorption efficiency of the adsorbents was slightly decreased after each adsorption and desorption cycle; however, compared with the first use, it still maintained a very high adsorption efficiency in the 15th use. According to the experimental data after 15 cycles of regeneration, the adsorption capacities of M 1.5 and N 1.5 for nitrate remained as 94.71% and 83.02% respectively, while for nitrite they remained as 97.38% and 81.15% respectively. These results confirm that the PEGDA-MTAC/NH_2_ adsorbents can be reused for nitrate and nitrite adsorption after the regeneration while reducing waste generation and providing a cost-effective application value for wastewater treatment.

### 3.3. Column Experiments

#### 3.3.1. Fixed Bed Adsorption/Desorption

Continuous flow adsorption experiments were performed in a column made of M 1.5 with an inner diameter of 20 mm and a height of 200 mm for nitrate- and nitrite-containing solutions. The initial concentration of 50 mg/L of nitrate and nitrite were passed in an upward direction through glass columns separately at a rate of 3 mL/min which were packed with 3.0 g of M 1.5. To avoid gravity-caused channeling, the solution was pumped upward. The breakthrough curves of nitrate and nitrite obtained are presented in Figure 10. In the case of nitrate, the effluents attained the breakthrough point at about 40 min. The nitrate adsorption capacities at the breakthrough point and the saturation point were 1.96 mg/g and 16.49 mg/g (influent time was 780 min) respectively. In the case of nitrite, the effluents attained the breakthrough point at about 60 min. The nitrite adsorption capacities at the breakthrough point and the saturation point were 2.88 mg/g and 17.54 mg/g (influent time was 840 min).

After completion of the M 1.5 column saturation, both nitrate and nitrite columns were washed with 500 mL of deionized water at a flow rate of 10 mL/min to remove excess nitrate and nitrite on the surface of the material. Next, the regeneration step was performed by 0.80 wt % of NaCl solution at a flow rate of 10.00 mL/min for 50 min. Then the columns were washed again with 500 mL of deionized water at a flow rate of 10 mL/min to remove the excess regenerant. Five cycles of adsorption/desorption were carried out and the results are shown in Figure 10 and Table 8 and Table 9. In both nitrate and nitrite cases, breakthrough points cannot be found from the second to fifth adsorption/desorption cycles. After the fifth cycle, the nitrate and nitrite adsorption capacities were respectively 10.52 mg/g (1080 min) and 6.61 mg/g (600 min) at saturation points, and the removal rate decreases as the number of regenerations increases. This is because the regeneration may be incomplete due to the fast flow rate and short residence time, resulting in nitrate remaining in the M 1.5 and entering the adsorption zone directly. These results indicated that the adsorption capacity of M 1.5 has been lost after regeneration. In addition, the slope of the curve becomes gradually steep after the regeneration. This phenomenon may account for the adsorption zone gradually narrowing [66], or rather the removal capacity for nitrate and nitrite gradually decreasing [69]. However, the efficiencies of nitrate and nitrite adsorption onto M 1.5 still remained as 46.05% and 52.81% respectively after being recycled five times, hence the column experiment is also applicable.

#### 3.3.2. Column Dynamic Study

The Thomas model was used to obtain a kinetic model in the column and estimate the breakthrough curves in order to describe the fixed-bed column behavior. The Thomas model was proposed by Henry C. Thomas in 1944 and it was predicated on the assumption that the process follows Langmuir adsorption–desorption kinetics with no axial dispersion. It states that the rate driving force is governed by second order reversible reaction kinetics [70]. The equation of the model (Equation (14)) and the linearized form of the equation (Equation (15)) are as follows:(14)$$\frac{C_{t}}{C_{0}}=\frac{1}{1+\exp(\frac{k_{\mathrm{Th}}q_{0}x}{Q}-k_{\mathrm{Th}}C_{0}t)}$$
(15)$$\ln\left(\frac{C_{0}}{C_{t}}-1\right)=\frac{k_{\mathrm{Th}}q_{0}x}{Q}-k_{\mathrm{Th}}C_{0}t$$
where k_Th_ is the Thomas model constant (mL/mg·min), q_o_ is the equilibrium adsorption amount of nitrate/nitrite on the adsorbent (mg/g), x is the mass of the adsorbent (g), and Q is the flow rate (mL/min). t is time (min). C_0_ and C_t_ are the initial concentration and the outflow concentration at a certain time (mg/L).

To determine the Thomas model constant (k_Th_) and equilibrium adsorption amount (q_0_), the experimental data were fitted with the Thomas model, and values were calculated from the slope and the intercepts of linear plots of ln (C_0_/C_t_ − 1) against t [71]. Thomas parameters and the linear regression coefficients (R^2^) are presented in Table 8 and Table 9, and Figure 11 shows the Thomas model fittings of each cycle of M 1.5 fixed-bed adsorption of nitrate and nitrite.

It can be observed that when the number of regenerations was increased, the values of the Thomas model constant (k_Th_) and the equilibrium adsorption amount (q_0_) were decreased, indicating that there were nitrate and nitrite residues on the adsorbent in each regeneration and this method cannot desorb nitrate and nitrite completely from M 1.5. The possible reasons for this observation are the flow rate of the regeneration step is too fast, the residence time of the desorbent on the surface of the adsorbent is short, and the chloride ion in the regeneration solution cannot effectively replace the nitrate and nitrite on the surface of the adsorbent.

## 4. Conclusions

In this work, hydrogel-based PEGDA-MCTA and PEGDA/AMHC were successfully synthesized through the photoreaction method and were used for the removal of nitrate and nitrite, applying both batch and continuous systems. When the initial pH of the solution increases from 2.0 to 12, the zeta potentials of PEGDA-MTAC and PEGDA/AMHC shifted from +6.91 to −13.12 mV and from +3.86 to −16.57 mV respectively. The adsorption capacity was increased significantly with the increasing modifier dose. The q_m_ values of M 1.5 and N 1.5 for adsorption of nitrate at 293 K were 13.66 mg/g and 11.25 mg/g respectively. Additionally, q_m_ values of M 1.5 and N 1.5 for adsorption of nitrite at 293 K were 13.30 mg/g and 13.12 mg/g respectively. The adsorption followed the Langmuir adsorption isotherms very well and the dynamic data coincided with the pseudo-second-order kinetic model. Nitrate and nitrite adsorption were decreased with the increasing temperature. The values of activation energies were laid between 25.13 and 29.81 kJ/mol for the reaction of nitrate and nitrite adsorption onto adsorbents used in this study indicate that the rate-determining step would be a diffusion-controlled process. The ΔH^0^ were positive; indicating endothermic adsorption. The negative values of ΔS^0^ express that the randomness was decreased at the solid-solution interfaces during nitrate and nitrite adsorption and the reaction was non-spontaneous. The positive values of ΔG^0^ express the non-spontaneous reaction for nitrate and nitrite adsorption onto M 1.5 and N 1.5. The adsorption of nitrate and nitrite onto M 1.5 and N 1.5 in high temperatures is difficult. Both composites achieved the desorption equilibrium within 15 min when the desorbent was NaCl. Adsorption efficiency of the adsorbents was slightly decreased after each adsorption and desorption cycle; however, compared with the first use, it still maintained a very high adsorption efficiency in the 15th use, proving that the PEGDA-MTAC/AMHC adsorbents have a practical application value. The continuous flow column studies indicated that the effluents attained to the breakthrough points of nitrate and nitrite were achieved at about 40 min and 60 min respectively. Based on these characteristics, the proposed hydrogel-based adsorbents can be more effective for nitrate and nitrite removal than most of the carbon-based adsorbents while these modified resins also reduce waste generation and can be used in a cost-effective way.

## Figures and Tables

**Figure 1 polymers-14-01442-f001:**
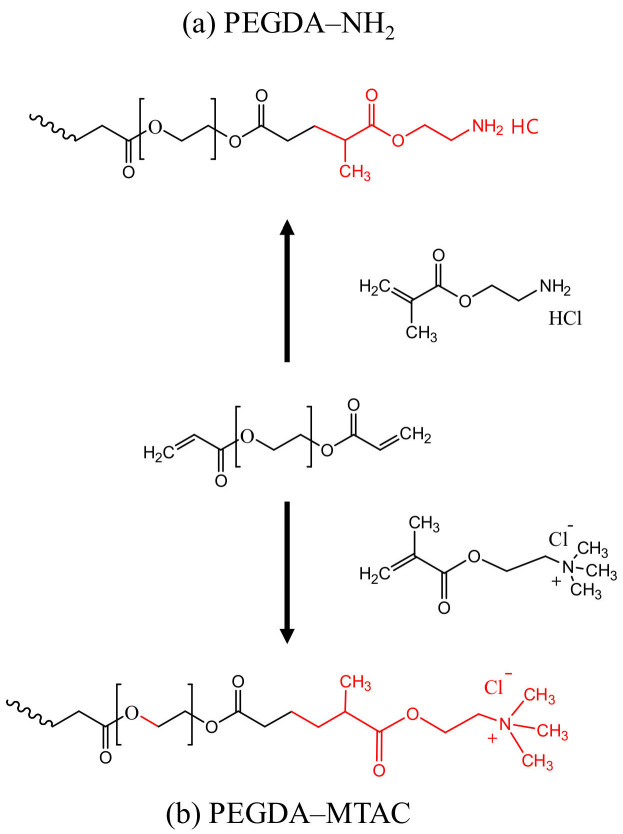
Synthesis of reusable hydrogel adsorbents (**a**) PEGDA–AMHC and (**b**) PEGDA–MTAC.

**Figure 2 polymers-14-01442-f002:**
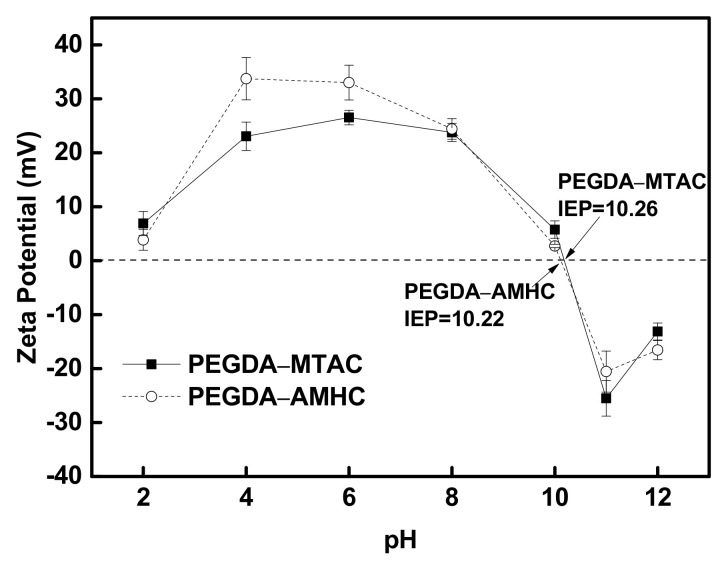
Zeta potentials of PEGDA–MTAC and PEGDA–AMHC.

**Figure 3 polymers-14-01442-f003:**
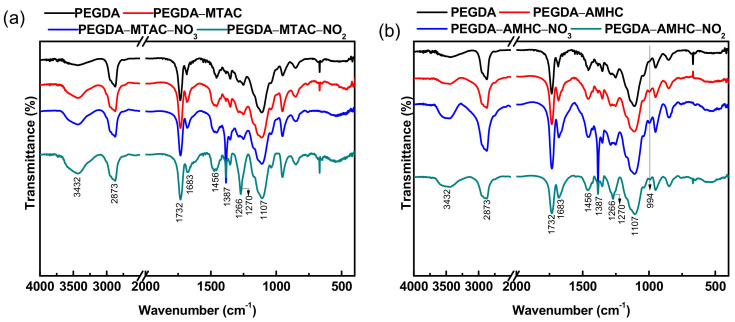
FT–IR spectra of (**a**) PEGDA–MTAC and (**b**) PEGDA–AMHC before and after adsorption of nitrate and nitrite.

**Figure 4 polymers-14-01442-f004:**
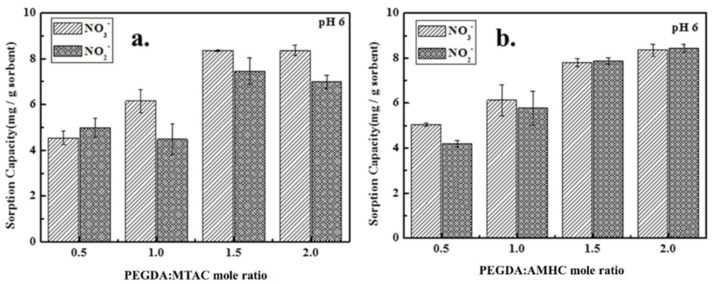
Concentration effect of modifier (MTAC or AMHC) toward adsorption capacity of nitrate and nitrite. (**a**) PEGDA–MTAC within 0.5, 1, 1.5, 2.0 (PEGDA: MTAC); (**b**) PEGDA–AMHC within 0.5, 1, 1.5, 2.0 (PEGDA: AMHC) (initial concentration of 50 mg N/L, pH = 6 and room temperature).

**Figure 5 polymers-14-01442-f005:**
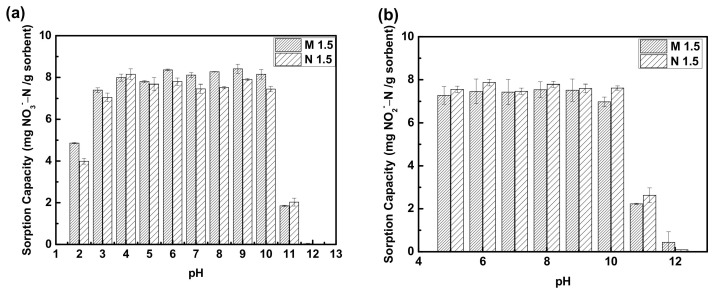
pH Effect (pH 2 to 12) for (**a**) nitrate and (**b**) nitrite adsorption using M 1.5 and N 1.5. Initial concentration of nitrate and nitrite were 50 mg N/L, adsorbent dose of 4 g/L at room temperature.

**Figure 6 polymers-14-01442-f006:**
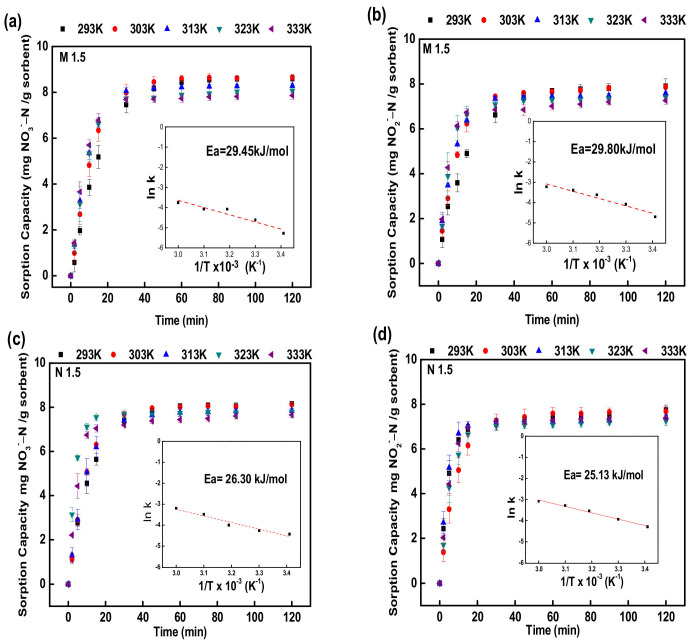
Adsorption capacities of nitrate and nitrite through M 1.5 and N 1.5 at different temperature. Adsorption capacity of nitrate (**a**) M 1.5 and (**c**) N 1.5; Adsorption capacities of nitrite (**b**) M 1.5 and (**d**) N 1.5. The inserted figures were fitting curves for activation energy of adsorption energy. Initial concentration of nitrate/nitrite was 50 mg N/L, adsorbent dose was 4 g/L, and pH = 5–6.

**Figure 7 polymers-14-01442-f007:**
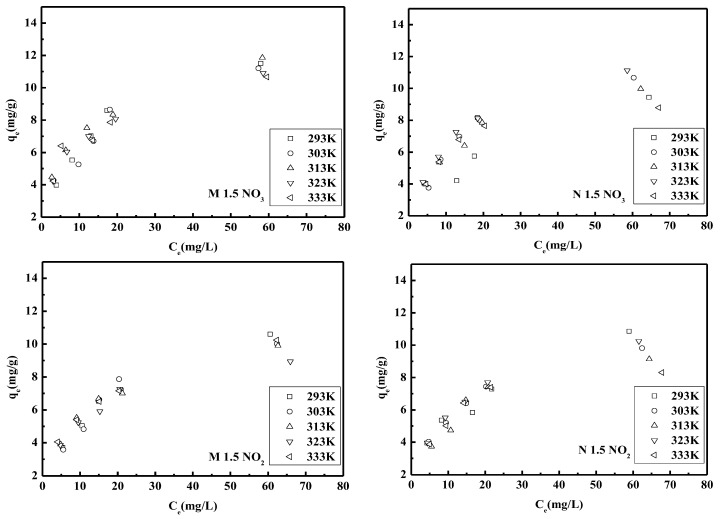
Adsorption isotherm of M 1.5 and N 1.5 at different temperatures (293–333 K). Adsorption capacity of nitrate for (**a**) M 1.5 and (**b**) N 1.5; adsorption of nitrite for (**c**) M 1.5 and (**d**) N 1.5 (dose of adsorbent = 4 g/L, pH = 5–6).

**Figure 8 polymers-14-01442-f008:**
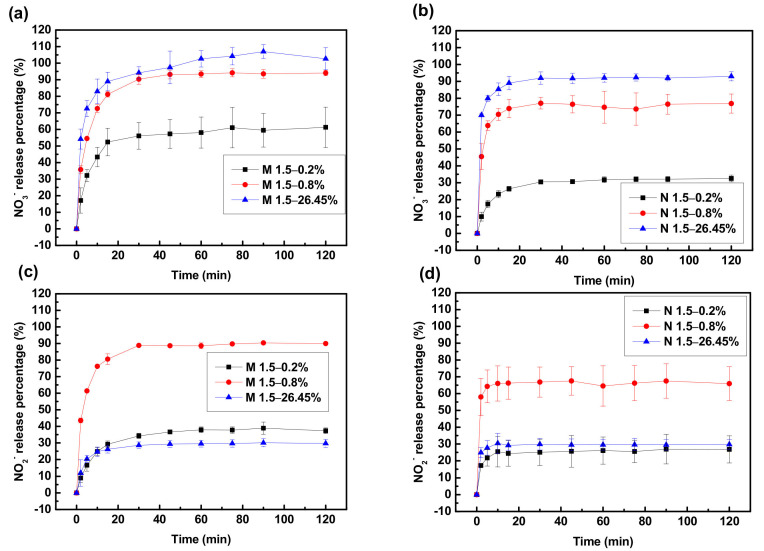
Time course of desorption of nitrate and nitrite from M 1.5 and N 1.5 using different concentrations (0.2 wt %, 0.8 wt % and 26.45 wt %) of NaCl solution. Nitrate releasing percentage (%) from (**a**) M 1.5; (**b**) N 1.5; Nitrite releasing percentage from (**c**) M 1.5; (**d**) N 1.5.

**Figure 9 polymers-14-01442-f009:**
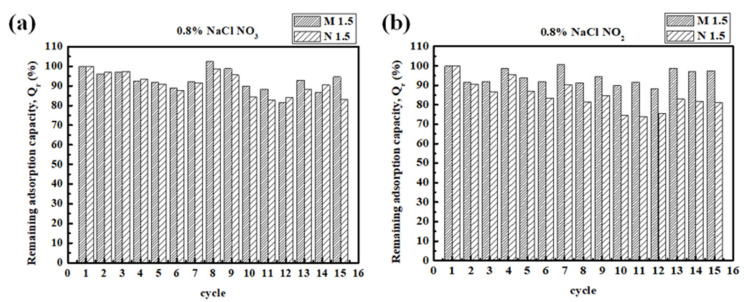
Regeneration of PEGDA-MTAC and PEGDA-AMHC using 0.80 wt % of NaCl in 15 cycles of adsorption/desorption (**a**) the remaining adsorption capacity of nitrate; (**b**) the remaining adsorption capacity of nitrite.

**Figure 10 polymers-14-01442-f010:**
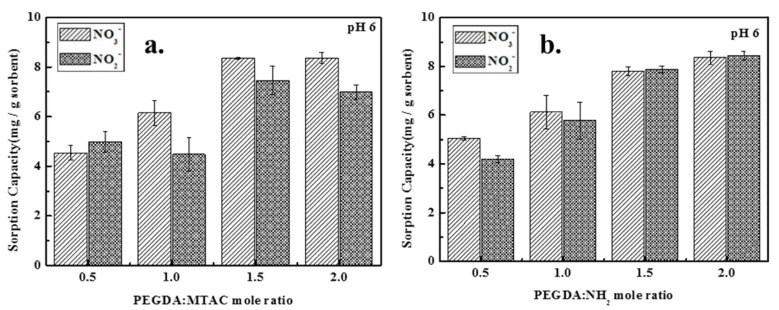
Fixed bed experiment conducted for adsorption of (**a**) nitrite and (**b**) nitrite on to M 1.5. (flow rate = 3 mL/min, column packed with 3.0 g of M 1.5, at room temperature). Regeneration was conducted by 0.80 wt % NaCl solution at a flow rate of 10.00 mL/min for 50 min.

**Figure 11 polymers-14-01442-f011:**
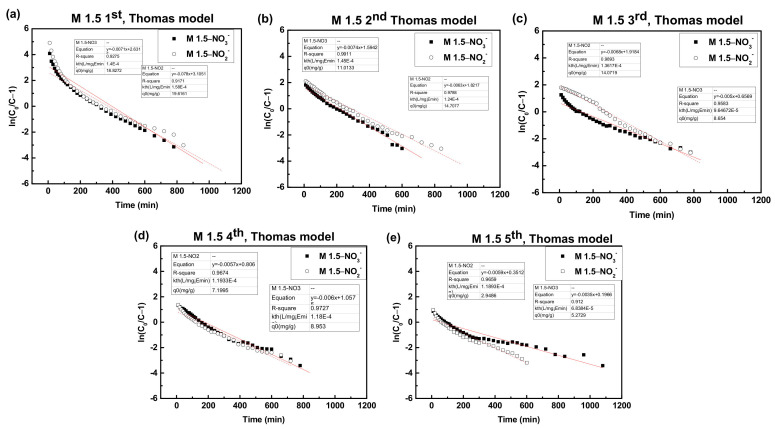
Thomas model fittings of nitrate and nitrite adsorption onto M 1.5 fixed bed column (**a**) first cycle, (**b**) second cycle, (**c**) third cycle (**d**) fourth cycle (**e**) fifth cycle. (flow rate = 3 mL/min, column packed with 3.0 g of M 1.5, at room temperature). Regeneration was conducted by 0.80 wt % NaCl solution at a flow rate of 10.00 mL/min for 50 min.

**Table 1 polymers-14-01442-t001:** FTIR analyses and peak assignments of both unloaded and loaded hydrogels.

Wave Numbers/cm^−1^	Group	Compound
3432	O–H stretching	Internal water
2873	C–H stretching	PEGDA, PEGDA-MTAC/AMHC
1732, 1683	C=O stretching	PEGDA, PEGDA-MTAC/AMHC
1107	C–O stretching	PEGDA, PEGDA-MTAC/AMHC
1466	C–H bending	PEGDA-MTAC/AMHC
1266	C–N stretching	PEGDA-MTAC/AMHC
1387	N–O stretching	PEGDA-MTAC/AMHC

**Table 2 polymers-14-01442-t002:** Adsorption kinetic parameters of the adsorption of nitrate onto M 1.5 and N 1.5 in aqueous solution.

Parameters	PEGDA-MTAC 1:1.5					PEGDA-AMHC 1:1.5				
	293 K	303 K	313 K	323 K	333 K	293 K	303 K	313 K	323 K	333 K
Pseudo-first-order										
q_e_ (mg/g)	9.1798	7.4231	5.2788	5.0113	4.0084	6.1159	5.0496	5.3431	1.9422	3.0606
k_1_ (min^−1^)	0.1391	0.1734	0.1621	0.1278	0.1340	0.1313	0.1317	0.1435	0.1361	0.1066
R^2^	0.9721	0.9489	0.9462	0.9388	0.8964	0.9389	0.8956	0.9642	0.8001	0.8561
Pseudo-second-order										
q_e_ (mg/g)	9.9108	9.6993	8.9206	8.6505	8.3056	8.9847	8.8417	8.4746	7.9114	7.8493
k_2_ (g/mg·min)	0.0051	0.0099	0.0170	0.0171	0.0234	0.0119	0.0141	0.0166	0.0307	0.0410
R^2^	0.9889	0.9917	0.9962	0.9969	0.9974	0.9968	0.9948	0.9969	0.9997	0.9994
Intraparticle diffusion										
k (mg/g·min^0.5^)	0.8558	0.8490	0.7626	0.7334	0.6851	0.7836	0.7632	0.7257	0.5404	0.5876
R^2^	0.9077	0.8286	0.7786	0.7781	0.7368	0.8471	0.7978	0.7965	0.5761	0.6555

**Table 3 polymers-14-01442-t003:** Adsorption kinetic parameters of the adsorption of nitrite onto M 1.5 and N 1.5 in aqueous solution.

Parameters	PEGDA-MTAC 1:1.5					PEGDA-AMHC 1:1.5				
	293 K	303 K	313 K	323 K	333 K	293 K	303 K	313 K	323 K	333 K
Pseudo-first-order										
q_e_ (mg/g)	6.6559	5.0455	3.6433	3.2275	2.9928	3.2847	4.9833	2.0333	2.9503	2.6018
k_1_ (min^−1^)	0.1172	0.1218	0.1105	0.1195	0.1011	0.1690	0.1382	0.0815	0.1117	0.1177
R^2^	0.9782	0.9355	0.8396	0.8742	0.8461	0.9885	0.9587	0.6610	0.8445	0.8187
Pseudo-second-order										
q_e_ (mg/g)	8.9206	8.4459	7.9365	7.6923	7.4571	8.3963	8.2034	7.5019	7.4963	7.5988
k_2_ (g/mg·min)	0.0091	0.0168	0.0269	0.0335	0.0398	0.0137	0.0195	0.0289	0.0375	0.0460
R^2^	0.9969	0.9977	0.9989	0.9986	0.9993	0.9958	0.9980	0.9995	0.9988	0.9991
Intraparticle diffusion										
k (mg/g·min^0.5^)	0.7822	0.7214	0.6478	0.6077	0.5608	0.7311	0.6911	0.5222	0.5812	0.5750
R^2^	0.8905	0.8017	0.7562	0.6940	0.6635	0.8244	0.7871	0.5953	0.6805	0.6527

**Table 4 polymers-14-01442-t004:** Activation energy parameters for nitrate and nitrite adsorption.

Parameters	PEGDA-MTAC 1:1.5		PEGDA-AMHC 1:1.5	
	NO_3_^−^	NO_2_^−^	NO_3_^−^	NO_2_^−^
E_a_ (kJ/mol)	29.4507	29.8090	26.2979	25.1312
A (g/mg·min)	1077.0704	2151.6709	517.2412	564.8160
R^2^	0.9148	0.9512	0.9405	0.9879

**Table 5 polymers-14-01442-t005:** Langmuir isotherm and thermodynamic parameters for adsorption nitrate by M 1.5 and N 1.5.

Parameters	PEGDA-MTAC 1:1.5					PEGDA-AMHC 1:1.5				
	293 K	303 K	313 K	323 K	333 K	293 K	303 K	313 K	323 K	333 K
Langmuir										
q_m_ (mg/g)	13.6612	12.3916	13.2802	12.1951	11.7371	11.2486	12.7714	11.5075	12.8866	9.6154
K_L_ (L/mg)	0.0912	0.1039	0.1242	0.1290	0.1433	0.0769	0.0856	0.1023	0.1036	0.1659
R_L_	0.0988	0.0878	0.0745	0.0719	0.0652	0.1151	0.1046	0.0890	0.0880	0.0569
R^2^	0.9973	0.9818	0.9904	0.9937	0.9891	0.9348	0.9990	0.9972	0.9980	0.9995
Thermodynamics										
ΔG° (kJ/mol)	29.0577	29.7368	30.4159	31.0949	31.7740	29.6692	30.2033	30.7374	31.2716	31.8057
ΔH° (kJ/mol)	9.1612					14.0191				
ΔS° (kJ/K·mol)	−0.0679					−0.0534				
R^2^	0.9755					0.8541				

**Table 6 polymers-14-01442-t006:** Langmuir isotherm and thermodynamic parameters for adsorption nitrite by M 1.5 and N 1.5.

Parameters	PEGDA-MTAC 1:1.5					PEGDA-AMHC 1:1.5				
	293 K	303 K	313 K	323 K	333 K	293 K	303 K	313 K	323 K	333 K
Langmuir										
q_m_ (mg/g)	13.2979	12.3305	11.4811	10.1215	11.8624	13.1234	11.4025	10.6838	11.9332	9.1491
K_L_ (L/mg)	0.0627	0.0720	0.0935	0.1131	0.0920	0.0691	0.0953	0.0948	0.0938	0.1535
R_L_	0.1376	0.1220	0.0966	0.0812	0.0980	0.1264	0.0950	0.0954	0.0963	0.0612
R^2^	0.9949	0.9921	0.9949	0.9975	0.9913	0.9599	0.9975	0.9945	0.9970	0.9985
Thermodynamics										
ΔG° (kJ/mol)	29.8930	30.5705	31.2480	31.9255	32.6029	29.7188	30.2944	30.8700	31.4457	32.0213
ΔH° (kJ/mol)	10.0425					12.8526				
ΔS° (kJ/K·mol)	−0.0678					−0.0576				
R^2^	0.7002					0.7620				

**Table 7 polymers-14-01442-t007:** Comparison of nitrate adsorption capacity on PEGDA-MTAC/AMHC with other reported adsorbents.

Adsorbent/Ion Exchange Resin	Adsorption Capacity (mg/g)	Reference
ZnCl_2_ treated coconut granular activated carbon	10.2	[35]
Modified lignite granular activated carbon	10	[68]
Polyvinyl alcohol/chitosan	35	[1]
Calcined (Mg–Al) hydrotalcite	34.36	[61]
Untreated coconut granular activated carbon	1.7	[35]
HDTMA modified QLD-bentonite	12.83–14.76	[15]
Cross-linked and quaternized Chinese reed	7.55	[42]
Wheat straw charcoal	1.10	[9]
PEGDA-MTAC/AMHC	13.51, 13.16	This study

**Table 8 polymers-14-01442-t008:** Thomas model parameters linear regression analysis for nitrate.

Cycle	PEGDA-MTAC 1:1.5						
	Breakthrough Point Time (min)	Operating Limit Time (min)	q_e(exp)_ (mg/g)	Removal (%)	Thomas Model		
					k_th_ (L/mg·min)	q_0_ (mg/g)	R^2^
1	40	780	16.49	42.34	1.63 × 10^−4^	17.78	0.9602
2	-	600	11.28	37.64	1.48 × 10^−4^	11.00	0.9914
3	-	780	9.87	25.33	1.00 × 10^−4^	6.91	0.9642
4	-	780	10.44	26.80	1.18 × 10^−4^	8.92	0.9676
5	-	1080	10.52	19.50	6.84 × 10^−5^	2.87	0.9120

**Table 9 polymers-14-01442-t009:** Thomas model parameters linear regression analysis for nitrite.

Cycle	PEGDA-MTAC 1:1.5						
	Breakthrough Point Time (min)	Operating Limit Time (min)	q_e(exp)_ (mg/g)	Removal (%)	Thomas Model		
					k_th_ (L/mg·min)	q_0_ (mg/g)	R^2^
1	60	840	17.54	41.75	1.62 × 10^−4^	19.36	0.9070
2	-	840	14.46	34.46	1.28 × 10^−4^	14.59	0.9808
3	-	780	14.06	35.83	1.38 × 10^−4^	13.97	0.9882
4	-	720	9.37	26.07	1.16 × 10^−4^	7.25	0.9620
5	-	600	6.61	22.05	1.19 × 10^−4^	2.95	0.9533

## Data Availability

The data presented in this study are available on request from the corresponding author.

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
