# Peer review of "Facile Synthesis of Hydrogel-Based Ion-Exchange Resins for Nitrite/Nitrate Removal and Studies of Adsorption Behavior"

_polymers, 2022, doi:10.3390/polym14071442_

Round 1

Reviewer 1 Report

To

The Editor,

Greetings of the day!

The present manuscript (Manuscript No.: polymers-1643611-peer-review-v1) has reported facile synthesis of low-cost recyclable hydrogel based resinous adsorbent suitable for removals of nitrate and nitrite. Herein, the authors have selected poly (ethylene glycol) diacrylate as the base material eventually modified by methacryloxyethyltrimethyl ammonium chloride (MTAC) and 2-Aminoethyl methacrylate hydrochloride (NH2) to incorporate trimethyl amine and amine functionalities, respectively. Adsorptive removals by poly (ethylene glycol) diacrylate type functionalized hydrogel have been reported earlier, however, this manuscript still contains satisfactory level of novelty. At the same time, the quality of technical writing needs improvement to remove the grammatical errors. As a whole, the manuscript also requires extensive revision to bring more clarity in the technical content. In my opinion, the manuscript should mandatorily be revised based on the following points:

  1. In the abstract, authors should insert full forms of both PEGDA-MTAC and PEGDA-NH2.

  1. In the first para of introduction, authors have mentioned demerits of only nitrate. However, the effects of nitrite contamination has not been covered.

  1. In the fourth para of the introduction, an order shows the maximum affinity of nitrate compared to sulfate, chloride, and bicarbonate. However, nothing has been shown regarding the affinity status of nitrite. Authors should provide this information collected from other works.

  1. If poly (ethylene glycol) diacrylate is a photodegradable material, such material is definitely not suitable for preparing sustainable and durable material. Authors should justify their choice.

  1. Authors should mention strategies behind modifications of PEGDA by amine and trimethyl amine in the penultimate para of introduction.

  1. It is not customary to abbreviate 2-Aminoethyl methacrylate hydrochloride by NH2, as NH2 is specifically used to represent amino. In this regard, I have no objection in using MTAC to abbreviate methacryloxyethyltrimethyl ammonium chloride.

  1. What is the meaning of ‘W’ in equation 1?

  1. Regeneration of hydrogel has been carried using brine solutions of particular concentration, such as 26.45 wt%. Is there any reason behind selecting such a particular concentration? Moreover, there is a huge gap between those selected concentrations – any reasons?

  1. In the FTIR analyses, peak at 1683 cm-1 designated as C–H bending seems inappropriate. Authors should reinvestigate this peak alongside providing suitable reference(s). Authors should also substantiate their claim regarding the peak at 1475 cm-1 by suitable reference(s).

  1. For better understanding of the readers, FTIR analyses and peak assignments of both unloaded and loaded hydrogels should be represented in a tabular form.

  1. While discussing the effect of pH, authors should reconnect the drastic drop in adsorption beyond pH 10 with the isoelectric points of the adsorbents. For isoelectric point/ zeta potential related change in adsorption profile, authors can go through the following papers: Journal of Molecular Liquids 293 (2019) 111470; Applied Surface Science 344 (2015) 188-195; Journal of Environmental Chemical Engineering 6 (2018) 289–310.

Finally, I hope that the authors will take suitable measures based on the abovementioned points to modify their manuscript before resubmission.

Author Response

Dear Editors,

Thank you for giving the opportunity to submit a revised draft of my manuscript titled “Facile Synthesis of Hydrogel-based Ion-Exchange Resins for Nitrite/Nitrate Removal and Studies of Adsorption Behavior” to Polymers. We appreciate that time and effort that you and the reviewers have dedicated to providing your valuable feedback on this manuscript. We are grateful to the reviewers for their insightful comments on our paper. We have been able to incorporate changes to reflect most of the suggestions provided by the reviewers. We have highlighted the changes within the manuscript.

Here is a point-to-point response to the reviewers’ comments and concerns.

Comment from Review 1

  1. In the abstract, authors should insert full forms of both PEGDA-MTAC and PEGDA-NH2.

Response: Corrected in the abstract and shown in yellow color.

  1. In the first para of introduction, authors have mentioned demerits of only nitrate. However, the effects of nitrite contamination have not been covered.

Response: Effects of nitrite contamination have added to the revised manuscript and colored in yellow color in the first paragraph of the introduction.

  1. In the fourth para of the introduction, an order shows the maximum affinity of nitrate compared to sulfate, chloride, and bicarbonate. However, nothing has been shown regarding the affinity status of nitrite. Authors should provide this information collected from other works.

Response: The affinity status of nitrite is added to the revised manuscript and colored in yellow color in the fourth paragraph of the introduction.

  1. If poly (ethylene glycol) diacrylate is a photodegradable material, such material is definitely not suitable for preparing sustainable and durable material. Authors should justify their choice.

Response: Polyethylene glycol (PEG) is a non-toxic, non-immunogenic, biocompatible, hydrophilic hydrogel and PEG derivatives, such as polyethylene glycol diacrylate, are most commonly functionalized with vinyl groups at the chain ends (PEGDA). PEGDA could photo-cross-link to form hydrogel adsorbents with biodegradable and high swelling characteristics that could be applied in recovery of heavy metals in waste waters and adsorption of dyes [1-3].

There is photodegradable poly (ethylene glycol)–based hydrogels. Example is nitrobenzyl ether-derived moiety, has been selected based on its photolytic efficiency for the modification of the poly (ethylene glycol)–based hydrogels polymer with photodegradable ability[4]. However, the PEGDA hydrogel catalysts which were used in our research are not having photodegradable functionality and this statement was modified and clearly mentioned with several cited papers.

  1. Authors should mention strategies behind modifications of PEGDA by amine and trimethyl amine in the penultimate para of introduction.

Response: Strategies behind modifications of PEGDA by amine and trimethyl amine were mentioned and colored in yellow color in the penultimate para of introduction.

  1. It is not customary to abbreviate 2-Aminoethyl methacrylate hydrochloride by NH2, as NH2 is specifically used to represent amino. In this regard, I have no objection in using MTAC to abbreviate methacryloxyethyltrimethyl ammonium chloride.

Response: The abbreviate 2-Aminoethyl methacrylate hydrochloride by NH2 was changed with AMHC (PEGDA_AMHC) and all the figures also changed relevant to that.

  1. What is the meaning of ‘W’ in equation 1?

Response: Meaning of ‘W’ has mentioned and colored in yellow color in equation 1.

  1. Regeneration of hydrogel has been carried using brine solutions of particular concentration, such as 26.45 wt%. Is there any reason behind selecting such a particular concentration? Moreover, there is a huge gap between those selected concentrations – any reasons?

Response: 26.45% of NaCl solution is the saturated concentration. Even though the spent PEGDA-MTAC can be regenerated under this concentration, the concentration of Cl ion would be potentially lead the secondary pollution due to the Cl release. Therefore, we collected low concentration, like 0.8 wt% of NaCl solution as regenerated reagent. The regeneration efficiency of spent PEGDA-MTAC and PEGDA-AHMC using low concentration of NaCl, especially 0.8wt% are close or even higher than using saturated NaCl solution. In addition, insufficient NaCl solution would not be strong enough to exchange adsorbed nitrate and nitrite on PEGDA-MTAC/PEGDA-AHMC adsorbents. Therefore, in this study, we plan to use low concentration of NaCl solution as regenerated reagent within a huge gap between selected concentration.

  1. In the FTIR analyses, peak at 1683 cm-1 designated as C–H bending seems inappropriate. Authors should reinvestigate this peak alongside providing suitable reference(s). Authors should also substantiate their claim regarding the peak at 1475 cm-1 by suitable reference(s).

Response: Peak at 1683 cm-1 was reinvestigated and corrected and the discussion on FT-IR section was rearranged with references in the manuscript and colored in yellow color.

  1. For better understanding of the readers, FTIR analyses and peak assignments of both unloaded and loaded hydrogels should be represented in a tabular form.

Response: FTIR analyses and peak assignments of both unloaded and loaded hydrogels have represented in a tabular form and colored in blue color. (Table 1).

  1. While discussing the effect of pH, authors should reconnect the drastic drop in adsorption beyond pH 10 with the isoelectric points of the adsorbents. For isoelectric point/ zeta potential related change in adsorption profile, authors can go through the following papers: Journal of Molecular Liquids 293 (2019) 111470; Applied Surface Science 344 (2015) 188-195; Journal of Environmental Chemical Engineering 6 (2018) 289–310.

Response: The effect of pH title, the drastic drop in nitrate and nitrite adsorption beyond pH 10 with the isoelectric points of the adsorbents were discussed with relevant references and colored in yellow color.

[5]

References

  1. Bhattacharyya, R. and S.K. Ray, Removal of congo red and methyl violet from water using nano clay filled composite hydrogels of poly acrylic acid and polyethylene glycol. Chemical Engineering Journal, 2015. 260: p. 269-283.
  2. Kwak, N.-S., et al., The effect of a molecular weight and an amount of PEGDA (poly(ethylene glycol)diacrylate) on a preparation of sodium methallyl sulfonate-co-PEGDA microspheres and sorption behavior of Co(II). Chemical Engineering Journal, 2013. 223: p. 216-223.
  3. Zhong, C., et al., Synthesis, characterization and cytotoxicity of photo-crosslinked maleic chitosan–polyethylene glycol diacrylate hybrid hydrogels. Acta Biomaterialia, 2010. 6(10): p. 3908-3918.
  4. Kloxin, A.M., et al., Photodegradable hydrogels for dynamic tuning of physical and chemical properties. 2009. 324(5923): p. 59-63.

Reviewer 2 Report

Manuscript Number: polymers-1643611

Title: Facile Synthesis of Hydrogel-based Ion-Exchange Resins for Nitrite/Nitrate Removal and Studies of Adsorption Behavior

Article Type: Article

In the manuscript the experimental research concerning adsorption properties of PEGDA-MTAC and PEGDA-NH2 hydrogels are presented. It has to be emphasized that the synthesis part of research is presented on just one page of the manuscript. The rest of the work is dedicated solely to the adsorption phenomenon. Therefore in my opinion the title of the manuscript is misleading and should be changed. Authors of the manuscript published in 2021 two other works dedicated the adsorption process. But in that cases other pollutants were removed. The topic of the research is interesting and worth publishing however I have some doubts if the manuscript fits the scope of Polymers journal.

The English of the work should be enhanced. Some sentences need to be corrected. I have found some minor errors. The strongest side of manuscript is extensive mathematical modelling of the adsorption process. The weakest side of the manuscript is a very short and inaccurate description of the synthesis process. Below I am presenting my remarks:

  1. The English should be improved. For example, it is “WHO have announced” it should be “WHO has announced” etc.
  2. Page 1 and further: Please correct “NO3-“
  3. Page 2: What kind of molecular weight? “Mn” or “Mw”? Give the unit in the manuscript. Did Authors determined the dispersity of polymer?
  4. ZetaPlus is able to determine zeta potential for particles up to 30 um in diameter (depending on the density). Please describe in details how the zeta potential was measured.
  5. Figure 2: Please add the standard deviation error bars.
  6. Table 3: In the case of PEGDA-NH2 the coefficients of determinations are quite low. Could Authors discuss this issue in the manuscript?
  7. Tables 4 and 5: Could Authors discuss more deeply the results of mathematical modelling of adsorption? For example, what is the reason for such high discrepancy between the results in the case of nitrate adsorption?
  8. The results discussion should be enhanced. Comparison to other works in this field od science should be made.

Author Response

2022/03/25

Dear Editors,

Thank you for giving the opportunity to submit a revised draft of my manuscript titled “Facile Synthesis of Hydrogel-based Ion-Exchange Resins for Nitrite/Nitrate Removal and Studies of Adsorption Behavior” to Polymer. We appreciate that time and effort that you and the reviewers have dedicated to providing your valuable feedback on this manuscript. We are grateful to the reviewers for their insightful comments on our paper. We have been able to incorporate changes to reflect most of the suggestions provided by the reviewers. We have highlighted the changes within the manuscript.

Here is a point-to-point response to the reviewers’ comments and concerns.

Comment from Reviewer 2

  1. The English should be improved. For example, it is “WHO have announced” it should be “WHO has announced” etc.

Response: Thank you for pointing this out. This term in this manuscript was already corrected.

  1. Page 1 and further: Please correct “NO3-

Response: Thank you for pointing this out. This term in this manuscript was already corrected.

  1. Page 2: What kind of molecular weight? “Mn” or “Mw”? Give the unit in the manuscript. Did Authors determined the dispersity of polymer?

Response: Molecular weight of PEGDA is Mw=700 and was corrected in the manuscript and colored in yellow color. We did not determined the dispersity of the polymer and PEGDA is a commercial product and we found the PDI of PEGDA (Mw=700) is similar to 0.011 (Mw/Mn) when dissolving in DI water (5.11 w%) and tetramethylethylenediamine (3 w%) [5]. (Size distribution and polydispersity index (PDI) for particles were determined by DLS using Zetasizer Nano ZS, ZEN3600, with a 4 mW and 633 nm wavelength He-Ne laser, measuring range being of 0.6 nm-6μm).

  1. Zeta-Plus is able to determine zeta potential for particles up to 30 um in diameter (depending on the density). Please describe in details how the zeta potential was measured.

Response:  Thank you for pointing this out. Generally, particles with diameters from 10nm to 30μm (depending on particle density) can be measured. Methodology for the determining the zeta potential was described in the materials and method section with relevant reference and colored in yellow color.

  1. Figure 2: Please add the standard deviation error bars.

Response: Thank you for pointing this out. The error bar already added in Figure 2.

  1. Table 3: In the case of PEGDA-NH2 the coefficients of determinations are quite low. Could Authors discuss this issue in the manuscript?

Response: Thank you for pointing this out. We recalculated the thermodynamic experimental data and we found that the mistake for the calculation. We already corrected this and arranged figure 6 and table 4 colored in yellow color.

  1. Tables 4 and 5: Could Authors discuss more deeply the results of mathematical modelling of adsorption? For example, what is the reason for such high discrepancy between the results in the case of nitrate adsorption?

Response: The results of the modelling of nitrate and nitrite adsorption on to PEGDA-MTAC/AMHC were discussed more deeply with several references and colored in yellow color.

  1. The results discussion should be enhanced. Comparison to other works in this field of science should be made.

Response: The results and discussion was enhanced while comparison this study with previous works and used both tabular form (Table 7) and description. This is also colored in yellow color.

References

  1. Radu, A.L., et al., Poly (ethylene Glycol) diacrylate-nanogels synthesized by mini-emulsion polymerization. 2019. 56: p. 514-519.

Round 2

Reviewer 1 Report

The present form of the manuscript appears to be improved considerably . Accordingly, I recommend acceptance of the manuscript in its present form.